# Take a Closer Look at Multilinguality! Improve Multilingual Pre-Training Using Monolingual Corpora Only

**Jinliang Lu**[1,2], **Yu Lu**[1,2], and **Jiajun Zhang**[1,2,3] [*]

[1]Institute of Automation, Chinese Academy of Sciences, Beijing, China
[2]School of Artificial Intelligence, University of Chinese Academy of Sciences, Beijing, China
[3]Wuhan AI Research, Wuhan, China
{jinliang.lu, yu.lu, jjzhang}@nlpr.ia.ac.cn

## Abstract

Recent studies have revealed the remarkable cross-lingual capability of multilingual pre-trained language models (mPLMs), even when pre-trained without parallel corpora (mono-mPLMs). Intuitively, semantic alignments may be the reason behind such capability but remain under-explored. In this work, we investigate the alignment properties from the token perspective in mono-mPLMs and find that the alignments correspond to the geometric similarity of embedding space across different languages. Nevertheless, mono-mPLMs tend to damage this geometric similarity at the higher layers due to the lack of cross-lingual interactions, thus limiting their cross-lingual transfer capabilities. To address this issue, we introduce token-level and semantic-level code-switched masked language modeling, employing the self-induced token alignments to explicitly improve cross-lingual interactions over layers of mono-mPLMs without relying on parallel sentences. We evaluate our method on various natural language understanding tasks and unsupervised machine translation tasks. The results demonstrate that our methods outperform the strong baselines and achieve comparable performance with mPLMs trained with parallel corpora.[1]

## 1 Introduction

Recent studies show that multilingual pre-trained language models (mPLMs) significantly improve the performance of cross-lingual natural language processing tasks. Conventional mPLMs (Devlin et al., 2019; Conneau and Lample, 2019; Conneau et al., 2020; Xue et al., 2021) typically adopt multiple monolingual corpora (mono-mPLMs) to perform masked language modeling during pre-training, obtaining impressive and stable multilingual capabilities, which may intuitively result

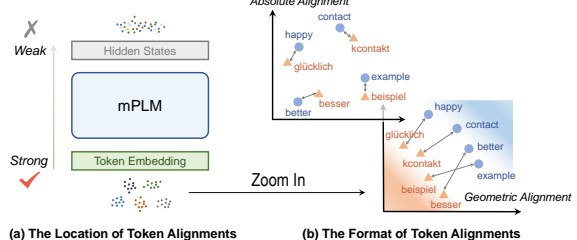

Figure 1: Illustration of properties of token alignments.

from the semantic alignments but remains under-explored. Another line of research involves improving the multilingual pre-training by incorporating the cross-lingual parallel corpora (para-mPLMs) into pre-training (Conneau and Lample, 2019; Cao et al., 2020; Chi et al., 2021a,b,c; Luo et al., 2021; Wei et al., 2021; Ouyang et al., 2021). However, parallel sentences are not always available, especially for low-resource languages (Tran et al., 2020). And collecting such data often entails substantial costs. Therefore, exploring approaches to improve multilingual pre-training without using parallel corpora is important and worthy of study.

To achieve this, we first conduct analyses to investigate the token alignment properties of XLM-R, a strong mono-mPLM that only uses multiple monolingual corpora for pre-training. Our empirical study demonstrates that the cross-lingual token alignments occur at the embedding layer (*alignment location*) with surprisingly high alignment accuracy (*alignment degree*) but they become weaker at the higher layers (Figure 1 (a)). We also find that the alignments are geometrically aligned instead of absolutely aligned (*alignment format*), as shown in Figure 1 (b). The phenomenon shows that token embeddings of different languages are separately distributed but geometrically similar.

We further compare the differences in geometric similarities of representations from the bottom layer to the top layer using mono-mPLMs and para-mPLMs. And we find that the representations be-

---

[*]Corresponding author
[1]Our code is available in https://github.com/JinliangLu96/Self-Improving-Multilingual-PT

come geometrically dissimilar at higher layers of mono-mPLMs while para-mPLMs would alleviate the problem by using parallel sentences, obtaining better cross-lingual transfer capability. It shows the necessity of explicit cross-lingual interactions.

Based on the above observation, we propose self-improving methods to encourage cross-lingual interactions using self-induced token alignments. Intuitively, the masked tokens can be predicted with semantic-equivalent but slightly language-mixed contexts. Therefore, we first utilize self-induced alignments to perform token-level code-switched masked language modeling (TCS-MLM), which requests the model to predict original masked tokens with the semantic-equivalent but code-switched surrounding text. Considering that vanilla replacements usually lack diversity, we further propose a novel semantic-level code-switched masked language modeling (SCS-MLM), which replaces the context tokens with a weighted combination of multiple semantically similar ones in other languages. SCS-MLM involves *on-the-fly* semantic replacements during training, further enhancing the diversity of code-switched examples and cross-lingual interactions.

We evaluate our methods on various cross-lingual transfer tasks. Specifically, we conduct experiments on natural language understanding tasks, including XNLI (Conneau et al., 2018) and PAWS-X (Hu et al., 2020) for sentence-pair classification, Wikiann (Pan et al., 2017) and UDPOS (Nivre et al., 2018) for structural prediction, MLQA (Lewis et al., 2020) for question answering, and Tatoeba (Artetxe and Schwenk, 2019) for sentence retrieval. We also perform experiments on unsupervised machine translation to evaluate the performance of the generation task. Experimental results demonstrate that our methods significantly improve the performance compared with the strong baselines, even surpassing some mPLMs pre-trained using parallel corpora. Further analysis demonstrates that our methods improve the geometric similarity of representations for different languages, and thus promoting the cross-lingual transfer capability.

Our contributions are summarized as follows:

- Our empirical study shows the existence of cross-lingual token alignments in mono-mPLMs. We further measure their accuracy, identify the location, and verify the format.

- Comparing mono-mPLMs with para-mPLMs, we find that mono-mPLMs tend to disturb

the geometric similarities between representations at higher layers while para-mPLMs remain unaffected, showing the necessity of cross-lingual interactions during pre-training.

- We propose token-level/semantic-level code-switched masked language modeling to encourage cross-lingual interactions during pre-training, improving the cross-lingual transfer capability without relying on parallel corpora.

## 2 A Closer Look at Multilinguality

In this section, we take the commonly used XLM-R, the strong mono-mPLM, as an example to show our observation[2]. Specifically, we first investigate the properties of cross-lingual token alignments in mono-mPLMs, showing their relation to geometric similarity. Then, we explore the variation of geometric similarity of representations across different layers and demonstrate that the geometric similarity at higher layers would be disturbed due to the lack of cross-lingual interactions, hindering the cross-lingual transfer capability.

### 2.1 Language-Specific Vocabulary

Generally, mPLMs adopt a huge vocabulary shared across 100+ languages. Different languages always have shared and independent tokens. Previous studies (Conneau and Lample, 2019; Pires et al., 2019; Wu and Dredze, 2019) regard the shared token as the source of cross-lingual capability. However, the latent relevance between language-specific tokens is not fully exploited.

Suppose that each language $l$ in the language set has a corresponding corpus $\mathcal{C}_l$. We first record the tokens whose frequencies are larger than 100 in $\mathcal{C}_l$, obtaining the vocabulary $\mathcal{V}_l$ for the specific language. Then, we remove shared tokens when processing two languages $l_a$ and $l_b$ to avoid the impact of overlapping. Finally, we obtain the language-specific vocabularies independent of each other:

$$\hat{\mathcal{V}}_{l_a} = \{t | t \in \mathcal{V}_{l_a} \text{ and } t \notin \mathcal{V}_{l_b}, \text{freq}(t) > 100\}$$
$$\hat{\mathcal{V}}_{l_b} = \{t | t \notin \mathcal{V}_{l_a} \text{ and } t \in \mathcal{V}_{l_b}, \text{freq}(t) > 100\}$$

### 2.2 Token-Alignments in Mono-mPLMs

After obtaining the language-specific vocabularies from other languages to English, we calculate the

---

[2]We also conduct analyses on mT5 (Encoder-Decoder) and X-GLM (Decoder). The phenomenon also exists regardless of the architecture, which is included in Appendix C.

| Models | params. | #lg | ar 1505 | bg 3322 | de 2187 | el 1223 | fr 2164 | hi 1159 | id 2381 | ja 2501 | ko 1278 | ru 4077 | Avg. |
|---|---|---|---|---|---|---|---|---|---|---|---|---|---|
| | | | | | | Hit@1 Accuracy | | | | | | | |
| XLM-R$_{\text{BASE}}$ | 250M | 100 | 44.12 | 58.61 | 48.33 | 52.58 | 49.03 | 48.92 | 47.00 | 62.38 | 63.77 | 59.43 | 53.42 |
| XLM-R$_{\text{LARGE}}$ | 560M | 100 | 46.71 | 58.79 | 48.61 | 52.25 | 48.94 | 50.13 | 47.08 | 63.97 | 62.52 | 59.97 | 53.90 |
| | | | | | | Hit@5 Accuracy | | | | | | | |
| XLM-R$_{\text{BASE}}$ | 250M | 100 | 68.04 | 72.91 | 60.86 | 65.09 | 59.10 | 60.74 | 56.82 | 75.37 | 72.77 | 75.55 | 66.72 |
| XLM-R$_{\text{LARGE}}$ | 560M | 100 | 69.77 | 73.81 | 59.95 | 66.48 | 57.95 | 61.78 | 56.82 | 77.09 | 73.94 | 76.21 | 67.38 |
| | | | | | | RSIM scores | | | | | | | |
| XLM-R$_{\text{BASE}}$ | 250M | 100 | 0.46 | 0.64 | 0.63 | 0.50 | 0.64 | 0.40 | 0.72 | 0.52 | 0.42 | 0.68 | 0.56 |
| XLM-R$_{\text{LARGE}}$ | 560M | 100 | 0.49 | 0.64 | 0.56 | 0.61 | 0.61 | 0.53 | 0.66 | 0.44 | 0.45 | 0.66 | 0.57 |

Table 1: Alignment accuracy/RSIM of translation pairs derived from different sizes of XLM-RoBERTa models across different languages to English. The number below the language is the size of exported cross-lingual dictionary.

similarity among token embeddings of XLM-R and directly export high-quality cross-lingual token alignments as dictionaries.

Specifically, we adopt the cross-domain similarity local scaling (CSLS) (Lample et al., 2018) to compute the token similarity from language *X* to language *Y*. For token embeddings *x* and *y* in two languages, the CSLS score is computed as:

$$\text{CSLS}(x,y) = 2\cos(x,y) - r_K(x) - r_K(y) \quad (1)$$

where $r_K(x)$ is the average score from $x$ to the $K$-nearest target neighbourhoods $\mathcal{N}(x)$. And $r_K(y)$ is vice versa.

$$r_K(x) = \frac{1}{K} \sum_{\hat{y}_t \in \mathcal{N}(x)} \cos(x, \hat{y}_t) \quad (2)$$

**Accuracy of Token-Alignments**   To measure the quality of dictionaries, we collect golden dictionaries from wiki-dict[3] and MUSE (Lample et al., 2018). The accuracy scores are shown in Table 1.

We find that the exported cross-lingual dictionaries have good quality, demonstrating that mono-mPLMs learn the alignments between different languages using monolingual corpora only. Particularly, distant language pairs, such as En-Ja (63.97%) and En-Ko (62.52%), have higher accuracy scores, while they usually have little overlapping tokens as anchor points for alignment. The phenomenon directly proves that the cross-lingual ability does not only depend on the overlapping tokens in different languages. Another potential factor could be the occurrence of words with similar meanings at comparable frequencies across languages (K et al., 2019). Using language modeling as the objective may unearth such regularities and stimulate the cross-lingual transfer capability.

**Format of Token-Alignments**   The second question is whether the alignments are *absolute alignment* or *geometrical alignment* (Figure 1 (b)). *Absolute alignment* requests the token embeddings are language-agnostic (Artetxe et al., 2017; Lample et al., 2018) while *geometrical alignment* focuses on the correspondence between tokens in different languages (Vulić et al., 2020). The latter just need to have a similar geometric spatial structure while keeping language characteristics (Roy et al., 2020).

Thus, we visualize the embedding of tokens in the exported dictionaries. The results across five diverse languages are shown in Figure 2 (a). We can find that the token embeddings are separately distributed in space according to the language similarity instead of aggregating together, showing that the alignments are *geometrical alignment*. We also use RSIM[4] to measure the geometric similarity in Table 1. By contrast, token representations at the top layer aggregate together (Figure 2 (b)).

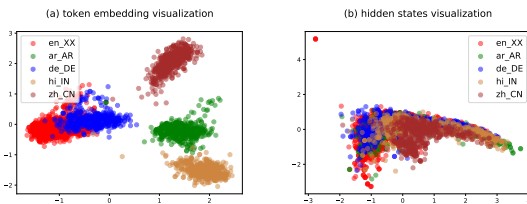

Figure 2: The visualization of embedding and states.

**Location of Token-Alignments**   Since top-layer hidden states aggregate, whether the accuracy of token alignments improves from the bottom to the top layers becomes a question. To answer the question, we compute the token alignment accuracy, average

---

[3] https://github.com/onny/wikidict

[4] RSIM means relational similarity, which measures the Pearson correlation coefficient between the cosine similarity matrices of intra-language tokens. It measures the degree of isomorphism (geometric similarity) of embedding spaces. And the calculation method is included in the appendix B.

cosine similarity, and RSIM scores using hidden states of different layers. Figure 3 (a) shows that with the layers becoming higher, token alignment accuracy decreases but the average cosine similarity between translation pairs increases, demonstrating that cross-lingual token alignments mainly exist in the embedding layer while higher layers focus on the aggregation of language-specific token representations. Moreover, Figure 3 (b) shows that the geometric similarities between language-specific token representations at top layers become weaker.

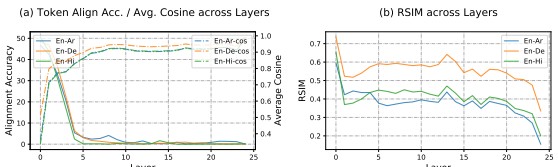

Figure 3: The token alignment accuracy/cosine similarity and RSIM across different layers of XLM-R$_{\text{LARGE}}$.

### 2.3 Cross-Lingual Interactions Matters for Geometric Similarity Maintenance

§2.2 shows that language-specific token representations of XLM-R at higher layers aggregate together (Figure 2 (b)) but the alignment and geometric similarity are disturbed (Figure 3). Since para-mPLMs usually obtain better performance on cross-lingual transfer tasks, we compare the difference between para-mPLMs and XLM-R (mono-mPLM) in the above aspects. Specifically, we choose VECO (Luo et al., 2021) and INFOXLM (Chi et al., 2021b) as representatives of para-mPLMs, which are pre-trained with monolingual and parallel corpora, obtaining improvements compared with XLM-R.

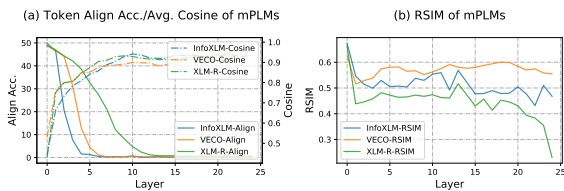

Figure 4: Comparison between XLM-R (mono-mPLM) and VECO/INFOXLM (para-mPLM). Each score is averaged by 3 language pairs: En-Ar, En-De, and En-Hi.

Figure 4 (a) shows the phenomenon of token alignment accuracy and cosine similarity across layers. We find that different mPLMs exhibit similar behavior, wherein both mono-mPLM and para-mPLMs tend to aggregate token representations while ignoring alignments at the higher layers. The reason behind this may lie in that higher layers of

PLMs prioritize complex semantic combinations rather than token features (Jawahar et al., 2019).

Figure 4 (b) compares the average RSIM scores of different mPLMs. VECO and INFOXLM have higher RSIM scores than XLM-R cross layers, showing that parallel corpora would improve the geometric similarity between languages. Furthermore, RSIM scores of VECO/INFOXLM across layers are more balanced than XLM-R. It demonstrates that explicit cross-lingual interactions (parallel corpora) are useful in maintaining geometric similarity in mPLMs, which could be one of the factors contributing to better cross-lingual transfer capability than mono-mPLMs.

## 3 Our Method

§2.2 demonstrates that mono-mPLMs learn cross-lingual token alignments and can export them as high-quality dictionaries. §2.3 shows explicit cross-lingual interactions may enhance cross-lingual transfer capability. These observations motivate us to explore self-improving methods to increase cross-lingual interactions without relying on parallel corpora. Next, we introduce our proposed token-level/semantic-level code-switch masked language modeling for multilingual pre-training.

### 3.1 Token-Level Code-Switch MLM (TCS)

Previous code-switch methods either rely on the existing bilingual dictionaries (Lin et al., 2020; Chaudhary et al., 2020) or the alignment tools to build the alignment pairs using parallel sentences (Ren et al., 2019; Yang et al., 2020). Our work proves that the self-induced dictionaries from mono-mPLMs are accurate enough, which would help mono-mPLMs self-improving and do not require the use of prior bilingual knowledge.

Therefore, we replace 10∼15% tokens using the dictionary to construct multilingual code-switched contexts but keep the masked positions unchanged, forcing the model to predict the masked tokens with different but semantic-equivalent contexts. For example, the original English token sequence (after cutting) is converted into a code-switch one using the self-induced dictionary (En-De):

$$\_A \ \_cat \ \_sit \ \_on \ \_the \ [mask] . \Rightarrow \mathcal{L}_{\text{MLM}}$$
$$\Downarrow$$
$$\_A \ \_\textbf{Katze} \ \_sit \ \_on \ \_the \ [mask] . \Rightarrow \mathcal{L}_{\text{TCS-MLM}}$$

Both the original and code-switched sentences are fed into mono-mPLMs to perform masked lan-

guage modeling. The training loss is:

$$\mathcal{L} = \mathcal{L}_{\text{MLM}} + \mathcal{L}_{\text{TCS-MLM}} \qquad (3)$$

## 3.2 Semantic-Level Code-Switch MLM (SCS)

Considering that token replacements often lack diversity, we propose a novel semantic-level code-switch method, which replaces 10∼15% tokens with the average weighting of its neighbors in another language, as shown in Figure 5.

Considering that mPLMs provide contextual output distributions across the vocabulary and avoid polysemy problems (Tversky and Gati, 1982), we first utilize the mono-mPLM to obtain the output probability distribution across the vocabulary for tokens. Then, we choose top-$k$[5] tokens according to probabilities and average-weight their embeddings as the contextual token representation $\hat{x}$:

$$\hat{\boldsymbol{x}} = \sum_{i=1}^{K} p_i \cdot \boldsymbol{e}_i \qquad (4)$$

where $p_i$ is the normalized probability of $i$-th token in the top-$k$ tokens.

After obtaining the contextual representations, we adopt the embedding table to search for corresponding translations *on-the-fly* instead of directly using the discrete dictionaries, which would improve the diversity of training examples and keep the semantics. Similarly, we also keep top-$k$ translation candidates and average-weighting their embedding as the replacement $\hat{y}$:

$$\hat{\boldsymbol{y}} = \sum_{j=1}^{K} q_j \cdot \boldsymbol{e}_j \qquad (5)$$

where $q_j$ is the normalized $\text{CSLS}_{\textit{on-the-fly}}$ scores[6] across the top-$k$ tokens in the corresponding language-specific vocabulary $\hat{\mathcal{V}}$.

$$q_j = \frac{\exp(\hat{q}_j)}{\sum_{m=1}^{K} \exp(\hat{q}_j)} \qquad (6)$$

$$\hat{q}_j = \text{CSLS}_{\textit{on-the-fly}}(\hat{\boldsymbol{x}}, y_j), \; y_j \in \hat{\mathcal{V}} \qquad (7)$$

Same as §3.1, we request the mono-mPLMs to perform masked language modeling based on the original examples and semantically code-switched ones. The final training loss is:

$$\mathcal{L} = \mathcal{L}_{\text{MLM}} + \mathcal{L}_{\text{SCS-MLM}} \qquad (8)$$

---

[5]$k$ is set as 8 in our experiments.
[6]During training, we compute $\text{CSLS}_{\textit{on-the-fly}}$ scores for tokens in the same batch to avoid expensive computational costs.

## 4 Pre-training Settings

**Pre-training Data**   We collect monolingual corpora from Common Crawl Corpus, which contains about 890GB data for 50 languages. Different from previous studies, we do not use any bilingual corpus. Following (Conneau and Lample, 2019), we sample multilingual data according to a multinomial distribution with probabilities. Considering the pre-training corpora in $N$ languages with $n_i$ training instances for the $i$-th language, the probability for $i$-th language can be calculated as:

$$p_i = \frac{n_i^{\alpha}}{\sum_{i=1}^{N} n_k^{\alpha}} \qquad (9)$$

where $\alpha$ is set as 0.7.

**Model Configuration**   Due to the restriction of resources, we conduct experiments on the Transformer encoder models to verify the effectiveness of our method. For fair comparisons with previous studies on natural language understanding tasks, we pre-train a 12-layer Transformer encoder as the BASE model (768 hidden dimensions and 12 attention heads) and a 24-layer Transformer encoder as the LARGE model (1024 hidden dimensions and 16 attention heads) using `fairseq` toolkit. The activation function used is GeLU. Following (Chi et al., 2021b; Luo et al., 2021; Ouyang et al., 2021), we initialize the parameters with XLM-R.

We also pre-train the 6-layer Transformer encoder (1024 hidden dimensions and 8 attention heads), which is adopted to evaluate the performance on unsupervised machine translation.

**Optimization Settings**   We use the Adam optimizer to train our model, whose learning rate is scheduled with a linear decay with 4000 warm-up steps. The peaking learning rates are separately set as 2e-4 and 1e-4 for BASE and LARGE model. Pre-training is conducted using 8 Nvidia A100-80GB GPUs with 2048 batch size. The BASE model takes about 1 month and the LARGE model takes about 2 months for pre-training. Appendix A shows more details about the pre-training settings.

## 5 Experiments on Downstream Tasks

### 5.1 Natural Language Understanding

#### 5.1.1 Experimental Settings

We consider four kinds of cross-lingual NLU tasks:

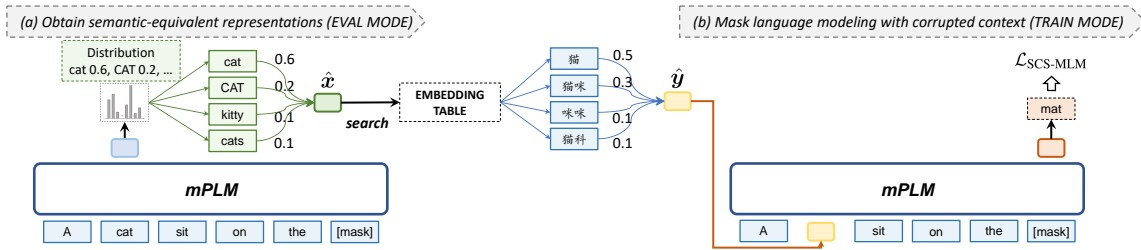

Figure 5: The illustration of SCS-MLM. First, we use the mPLM to predict the distribution of the context token, obtaining the contextual token representations and searching corresponding translations. Then, we use the weighted representations to replace the corresponding token and request the model to predict the same masked tokens.

**Sentence-Pair Classification** We choose XNLI (Conneau et al., 2018) for cross-lingual language inference and PAWS-X (Hu et al., 2020) for cross-lingual paraphrase identification.

**Structural Prediction** We choose UDPOS (Nivre et al., 2018) for pos-tagging and Wikiann (Pan et al., 2017) for name entity recognition.

**Question Answering** We choose MLQA (Lewis et al., 2020) for cross-lingual question answering.

**Cross-lingual Retrieval** We choose Tatoeba (Artetxe and Schwenk, 2019) for parallel sentence identification.

### 5.1.2 Experimental Results

| Model | XNLI | PAWSX |
|---|---|---|
| *Pre-training without parallel corpus* | | |
| mBERT (Devlin et al., 2019) | 65.4 | 81.9 |
| XLM (Conneau and Lample, 2019) | 69.1 | 80.9 |
| XLM-R$_{BASE}$ (*our-impl*) | 74.9 | 84.4 |
| + TCS-MLM | 75.2 | 84.9 |
| + SCS-MLM | **75.8** | **85.1** |
| XLM-R$_{LARGE}$ (Chen et al., 2022) | 79.2 | 86.4 |
| + TCS-MLM | 80.8 | 88.2 |
| + SCS-MLM | **81.1** | **88.9** |
| *Pre-training with parallel corpus* | | |
| XLM (Conneau and Lample, 2019) | 75.1 | - |
| Unicoder | 75.4 | - |
| VECO$_{LARGE}$ (Luo et al., 2021) | 79.9 | 88.7 |
| VECO$_{LARGE}$ 2.0 (Zhang et al., 2023) | 80.4 | 88.5 |
| HICTL$_{LARGE}$ (Wei et al., 2021) | 81.0 | 87.5 |
| ERNIE-M$_{LARGE}$ (Ouyang et al., 2021) | **82.0** | **89.5** |

Table 2: Evaluation results on XNLI and PAWS-X cross-lingual text classification tasks. We report the average accuracy across different language test sets, which are based on five runs with different random seeds.

We conduct experiments on the above cross-lingual NLU tasks to evaluate the cross-lingual transfer capability of our method: fine-tune the model with the English training set and evaluate

the foreign language test sets. We separately describe the results as follows:

| Model | POS | NER |
|---|---|---|
| *Pre-training without parallel corpus* | | |
| mBERT (Devlin et al., 2019) | 70.3 | 62.2 |
| XLM (Conneau and Lample, 2019) | 70.1 | 61.2 |
| XLM-R$_{BASE}$ (Conneau et al., 2020) | 73.4 | 61.9 |
| + TCS-MLM | 73.2 | **62.7** |
| + SCS-MLM | **73.8** | 62.6 |
| XLM-R$_{LARGE}$ (Conneau et al., 2020) | 73.8 | 65.4 |
| + TCS-MLM | 75.1 | 66.0 |
| + SCS-MLM | **76.0** | **66.6** |
| *Pre-training with parallel corpus* | | |
| VECO$_{LARGE}$ (Luo et al., 2021) | 75.1 | 65.7 |
| VECO$_{LARGE}$ 2.0 (Zhang et al., 2023) | 75.4 | **67.2** |
| HICTL$_{LARGE}$ (Wei et al., 2021) | 74.8 | 66.2 |

Table 3: Evaluation results on UDPOS and Wikiann cross-lingual structural prediction tasks. We report the average F1 scores, which are based on five runs with different random seeds.

**Sentence-Pair Classification** The cross-lingual natural language inference (XNLI) aims to determine the relationship between the two input sentences, entailment, neural or contradiction. And PAWS-X aims to judge whether the two sentences are paraphrases or not.

As shown in Table 2, our SCS-MLM$_{BASE}$ surpasses the baseline models including mBERT, XLM, XLM-R$_{BASE}$ and Unicoder. Moreover, our SCS-MLM$_{LARGE}$ obtains equivalent performance with some pre-trained models using parallel sentences, including VECO$_{LARGE}$ and HICTL$_{LARGE}$. In contrast, although TCS-MLM also obtains improvements, it is not as good as SCS-MLM. We suppose that the limited dictionaries would lead to insufficient cross-lingual interactions.

**Structural Prediction** Structural prediction task contains UDPOS and Wikiann. Given a sentence, UDPOS aims to label the pos-tagging for tokens

| Models | ar | de | en | es | hi | vi | zh | Avg |
|---|---|---|---|---|---|---|---|---|
| *Pre-training without parallel corpus* | | | | | | | | |
| mBERT | 45.7 / 29.8 | 57.9 / 44.3 | 77.7 / 65.2 | 64.3 / 46.6 | 43.8 / 29.7 | 57.1 / 38.6 | 57.5 / 37.3 | 57.7 / 41.6 |
| XLM | 54.8 / 36.3 | 62.2 / 47.6 | 74.9 / 62.4 | 68.0 / 49.8 | 48.8 / 27.3 | 61.4 / 41.8 | 61.1 / 39.6 | 61.6 / 43.5 |
| XLM-R$_{BASE}$ | 54.9 / 36.6 | 60.9 / 46.7 | 77.1 / 64.6 | 67.4 / 59.6 | 59.4 / 42.9 | 64.5 / 44.7 | 61.8 / 39.3 | 63.7 / 46.3 |
| + TCS-MLM | 57.6 / 38.5 | 63.2 / 48.7 | 80.1 / 66.9 | 68.1 / 50.2 | 62.7 / 44.8 | 68.0 / 47.3 | 63.5 / 40.0 | 66.2 / 48.1 |
| + SCS-MLM | **58.0 / 39.2** | **63.5 / 49.0** | **81.1 / 68.1** | **69.0 / 51.0** | **63.6 / 46.2** | **68.7 / 47.9** | **65.1 / 42.5** | **67.0 / 49.1** |
| XLM-R$_{LARGE}$ | 63.1 / 43.5 | 64.3 / 53.6 | 80.6 / 67.8 | 74.1 / 56.0 | 69.2 / 51.6 | 71.3 / 50.9 | 68.0 / 45.4 | 70.7 / 52.7 |
| + TCS-MLM | 66.4 / 46.8 | 70.1 / 55.1 | 84.3 / 71.4 | **74.5** / 56.4 | **71.7** / 53.5 | 73.8 / 52.6 | 70.2 / 46.9 | 73.0 / 54.7 |
| + SCS-MLM | **66.6 / 46.9** | **70.2 / 55.3** | **84.4 / 71.6** | **74.5** / **56.5** | 71.5 / **53.9** | **74.3 / 53.2** | 70.7 / **47.7** | **73.2 / 55.0** |
| *Pre-training with parallel corpus* | | | | | | | | |
| VECO$_{LARGE}$ | 65.0 / 44.6 | 69.8 / 54.6 | 83.5 / 70.6 | 74.1 / 56.6 | 70.6 / 53.1 | 74.0 / 52.9 | 62.1 / 37.0 | 71.7 / 53.2 |
| VECO$_{LARGE}$ 2.0 | **74.3 / 56.3** | 70.3 / 54.9 | 84.1 / 71.4 | 66.5 / 46.5 | 71.5 / 53.7 | 74.2 / 53.1 | 67.9 / 43.7 | 72.7 / 54.3 |
| H$_{ICTL LARGE}$ | - | - | - | - | - | - | - | 72.8 / 54.5 |
| E$_{RNIE}$-M$_{LARGE}$ | 67.4 / 47.2 | **70.8 / 55.9** | **84.4 / 71.5** | **74.8 / 56.6** | **72.6 / 54.7** | **75.0 / 53.7** | **71.1 / 47.5** | **73.7 / 55.3** |

Table 4: Evaluation results on MLQA cross-lingual question answering. We report the F1 / exact match (EM) scores. The results of TCS-MLM and SCS-MLM are averaged over five runs.

and Wikiann aims to identify name entities. We reported the average F1 score for each dataset.

Table 3 shows the results of our models. Compared with previous studies, our proposed SCS-MLM$_{LARGE}$ obtains the best results on UDPOS, achieving 76.0 F1 score. For Wikiann, our TCS-MLM$_{LARGE}$ and SCS-MLM$_{LARGE}$ also obtain significant improvements compared with the strong baseline XLM-R. We suppose that the induced dictionaries contain the relations of entities and pos-tagging across different languages, which promotes improvements in the structural prediction tasks.

**Cross-lingual Question Answering** MLQA aims to answer questions based on the given paragraph, which contains 7 languages.

The F1/EM scores are shown in Table 4. We can find that our proposed TCS-MLM and SCS-MLM are significantly better than the strong baseline XLM-R and even surpass some models pre-trained with parallel sentences, such as VECO, VECO 2.0 and H$_{ICTL}$. Although our methods cannot surpass E$_{RNIE}$-M$_{LARGE}$, they narrow the gaps between mPLMs training with or without parallel sentences, demonstrating the effectiveness of our methods.

**Cross-lingual Retrieval** To evaluate the cross-lingual sentence retrieval capability of our models, we choose a subset of the Tatoeba dataset (36 language pairs), which aims to identify the parallel sentence among 1000 candidates. Following previous studies, we used the averaged representation in the middle layer of different models (XLM-R$_{BASE}$, + TCS-MLM$_{BASE}$ and + SCS-MLM$_{BASE}$) to evaluate the retrieval task.

The results are shown in Figure 6. We can

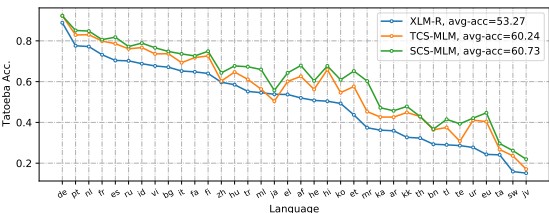

Figure 6: Tatoeba results for languages, which are sorted according to the performance of XLM-R$_{BASE}$.

find that our proposed SCS-MLM$_{BASE}$ obtains better retrieval accuracy scores (average acc. 60.73) than TCS-MLM$_{BASE}$ (+0.49 acc.) and XLM-R$_{BASE}$ (+7.46 acc.) across language directions, demonstrating the effectiveness of our method.

### 5.2 Natural Language Generation - UNMT

As our proposed pre-training methods do not rely on parallel sentences, we choose the harder task - unsupervised neural machine translation (UNMT) to evaluate the performance on the generation task.

#### 5.2.1 Experimental Results

Table 5 shows the translation performance on WMT14 En-Fr, WMT16 En-De, WMT16 En-Ro test sets. We can find that our proposed SCS-MLM can improve the translation quality compared with the strong baselines, XLM and MASS. For example, SCS-MLM respectively outperforms XLM and MASS by 1.1 and 1.2 BLEU scores in WMT16 En→Ro. SCS-MLM also surpasses previous studies on average, verifying its effectiveness. Moreover, the results also show that our method is suitable for the seq2seq model - MASS (Figure 8 in

| Pre-trained Models | En↔Fr | | En↔De | | En↔Ro | | Avg. |
|---|---|---|---|---|---|---|---|
| | En→Fr | Fr→En | En→De | De→En | En→Ro | Ro→En | |
| XLM | 33.4 | 33.3 | 26.4 | 34.3 | 33.3 | 31.8 | 32.1 |
| *our re-impl.* | 37.3 | 34.7 | 27.1 | 33.9 | 34.5 | 32.7 | 33.4 |
| Ren et al. (2019) | 35.4 | 34.9 | 27.7 | **35.6** | 34.9 | **34.1** | 33.8 |
| Ai and Fang (2022) | 34.1 | 34.0 | 27.2 | 34.9 | 34.2 | 32.6 | 32.8 |
| Ren et al. (2021) | 34.3 | 35.0 | **28.8** | 35.2 | 34.5 | 32.9 | 33.5 |
| XLM + SCS-MLM (*ours*) | **37.5** | **35.2** | 27.7 | 34.6 | **35.6** | 33.8 | **34.1** |
| MASS | 37.5 | 34.9 | **28.3** | 35.2 | 35.2 | 33.1 | 34.0 |
| *our re-impl.* | 37.1 | 34.7 | 27.4 | 34.9 | 34.9 | 33.2 | 33.7 |
| MASS + SCS-MASS (*ours*) | **37.9** | **35.4** | 27.9 | **35.8** | **36.1** | 33.7 | **34.5** |

Table 5: Unsupervised translation performance on WMT14 En-Fr, WMT16 En-De, WMT16 En-Ro. The results of previous studies are picked from corresponding papers.

appendix A.3), demonstrating that our method is independent of the model architectures.

### 5.3 SCS-MLM Improves Geometric Similarity at Higher Layers

§2.3 shows that para-mPLMs would maintain the geometric similarity of language-specific token representations across layers. As our method incorporate explicit cross-lingual interactions into pretraining, a similar phenomenon should occur.

Therefore, we plot the RSIM scores across 24-layer LARGE models for measurement. As shown in Figure 7 (a), compared with the baseline model, our proposed SCS-MLM increases the geometric similarity of different languages across layers. Figure 7 (b) shows the RSIM improvements focus on the higher layers, thereby achieving balanced geometric similarities, akin to the observations of para-mPLMs in Figure 4 (b). It could illustrate the reason why our method is effective on various cross-lingual transfer tasks.

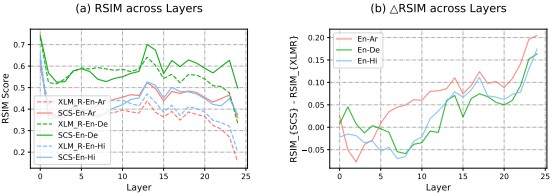

Figure 7: (a) RSIM scores across different layers of SCS-MLM_LARGE (solid lines) and XLM-R_LARGE (dashed lines). (b) The curve of $\Delta_{RSIM}$ across layers.

## 6 Related Work

Multilingual pre-trained language models begin with mBERT (Devlin et al., 2019) and XLM (Conneau and Lample, 2019), which learn the shared feature space among languages using multiple monolingual corpora. XLM-R (Conneau et al., 2020) shows the effects of models when trained on a large-scale corpus, establishing strong baselines for subsequent studies.

Based on the observation that parallel corpora would help cross-lingual alignment in (Conneau and Lample, 2019), many studies pay attention to the usage of the parallel corpus. Unicoder (Huang et al., 2019) employs a multi-task learning framework to learn cross-lingual semantic representations. ALM (Yang et al., 2020) and PARADISE (Reid and Artetxe, 2022) uses parallel sentences to construct code-switch sentences. INFOXLM (Chi et al., 2021b) and HICTL (Wei et al., 2021) respectively employ sentence-level and token-level contrastive learning for cross-lingual semantic alignments. VECO (Luo et al., 2021) proposes a variable framework to enable the model to process understanding and generation tasks. ERNIE-M (Ouyang et al., 2021) generates pseudo-training examples, further improving the overall performance on downstream tasks. (Ai and Fang, 2023) explores the MASK strategy (*prototype-word*) to improve cross-lingual pre-training.

Besides the above discriminative models, generative models, such as MASS (Song et al., 2019), mBART (Liu et al., 2020), mT5 (Xue et al., 2021), XGLM (Lin et al., 2022), BLOOM (Scao et al., 2022), also show impressive performance on generation tasks. MASS, mT5 and mBART pre-train the denoising seq2seq model, handling both NLU and NLG tasks. XGLM and BLOOM pre-train extremely large auto-regressive models using extremely large multiple monolingual corpora, showing surprising in-context learning (Brown et al.,

2020) capability on cross-lingual tasks.

Different from previous studies, our work first investigates the properties of token alignments behind multilinguality and then proposes self-improving methods for multilingual pre-training with only monolingual corpora, alleviating the need for parallel sentences.

# 7 Conclusion

In this work, we first investigate the properties of cross-lingual token alignments in mono-mPLMs, and then make a comparison between mono-mPLMs and para-mPLMs, demonstrating that geometric similarities of higher-layer representations would be damaged without explicit cross-lingual interactions, hindering the multilinguality. Therefore, we propose token-level and semantic-level code-switch masked language modeling to improve the cross-lingual interactions without relying on parallel corpora. Empirical results on language understanding and generation tasks demonstrate the effectiveness of our methods. Future work would adapt our methods to much larger language models.

# Acknowledgements

This work is supported by National Key R&D Program of China 2022ZD0160602 and the Natural Science Foundation of China 62122088.

# Limitations

In this paper, we investigate the token-level cross-lingual alignments but do not focus on the sentence-level alignments, which occur in the middle layer of mPLMs. We leave it as our future work. Then, we propose token-level and semantic-level code-switch masked language modeling for multilingual pre-training. As we analyzed in Appendix C, various types of mono-mPLMs are capable of forming alignments, but they are unable to preserve geometric similarities across layers. This phenomenon is observed irrespective of the language modeling methods employed. However, due to resource limitations, we only conduct experiments on XLM-R and MASS models. We will verify the effectiveness of our methods on much larger language models in the future, such as XGLM and BLOOM.

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

## A Experimental Settings

### A.1 Pre-Training Data

We use the open-source CC-100 corpora[7] for the pre-training of BASE and LARGE models. Due to the resource and memory restrictions, we just select 50 languages that cover the downstream tasks and conduct random sampling following(Luo et al., 2021). Table 6 shows the statistics of the monolingual data in each language.

| Code | Size (GB) | Code | Size (GB) |
|------|-----------|------|-----------|
| af | 2.07 | kk | 8.14 |
| ar | 10.29 | ko | 15.36 |
| bg | 22.75 | lt | 10.08 |
| bn | 10.5 | lv | 6.38 |
| cs | 19.87 | ml | 4.23 |
| de | 45.05 | mr | 3.45 |
| el | 27.96 | ms | 5.73 |
| en | 191.85 | my | 0.52 |
| es | 40.35 | ne | 4.55 |
| et | 9.17 | nl | 13.8 |
| eu | 2.95 | pl | 15.46 |
| fa | 26.63 | pt | 15.2 |
| fi | 19.84 | ro | 9.56 |
| fr | 44.16 | ru | 46.46 |
| fy | 0.27 | si | 4.5 |
| gd | 0.2 | sw | 2.56 |
| gu | 2.42 | ta | 7.26 |
| he | 7.01 | te | 5.66 |
| hi | 20.25 | tr | 14.05 |
| hu | 8.68 | th | 20.65 |
| id | 44.12 | tl | 4.84 |
| it | 9.03 | vi | 42.15 |
| ja | 18.86 | ur | 7.93 |
| jv | 0.15 | yo | 0.0006 |
| ka | 5.45 | zh | 40.14 |

Table 6: Statistic of pre-training data in our experiments.

### A.2 Hyperparameters for Pre-training

Table 7 shows the hyperparameters for pre-training different size models. SMALL models are initialized by XLM or MASS. And they are used to conduct experiments on unsupervised machine translation. BASE model and LARGE model are initialized by XLM-R model and used for various natural language understanding tasks.

---

[7] https://data.statmt.org/cc-100/

### A.3 Experimental Settings of UNMT

We follow the common practices to conduct experiments on UNMT benchmarks. For evaluation, we separately adopt *newsdev/test 2014* En-Fr, *newsdev/test 2016* En-De, *newsdev/test 2016* En-Ro as development and test sets.

For a fair comparison with previous studies, we pre-train the cross-lingual language models with the same model architecture of XLM and MASS. The pre-training data for UNMT is shown in Table 8. We compare SCS-MLM with other UNMT pre-training methods (Ren et al., 2019, 2021; Song et al., 2019; Ai and Fang, 2022), which have the equivalent number of parameters.

During inference, we use the beam size 1 and length penalty 1.0. To be consistent with previous works, we use multi-bleu.perl to measure the translation quality.

The illustration of our method on MASS is shown in Figure 8, which is similar to Figure 5 but needs to predict multiple adjacent tokens using the sequence-to-sequence model.

## B RSIM - Relational Similarity

In NLP, the geometric similarity between two embedding spaces can be measured by Relational Similarity (RSIM). Given $s$ translation pairs, we first calculate pairwise cosine similarities among intra-language tokens and obtain two vectors $a$ add $b$:

$$a = \cos(x_0, x_1), \cos(x_0, x_2), \cdots, \cos(x_s, x_{s-1})$$
$$b = \cos(y_0, y_1), \cos(y_0, y_2), \cdots, \cos(y_s, y_{s-1})$$

Then, we compute the Pearson's correlation between $a$ add $b$, which is known as Relational Similarity (Vulić et al., 2020).

## C Cross-Lingual Alignments on Different mPLMs

We also evaluate the accuracy of the token alignments across different language models. The results are shown in Table 9. Some examples are included in Figure 11.

We find that different kinds of language models form cross-lingual alignments. It demonstrates that pre-trained language models automatically learn cross-lingual mapping based on language modeling regardless of the specific modeling methods (i.e. masked language modeling, text span prediction, or casual language modeling). For different kinds

| Hyperparameters | SMALL | BASE | LARGE |
|---|---|---|---|
| Toolkit | XLM / MASS | fairseq / transformers | fairseq / transformers |
| Layers | 6 | 12 | 24 |
| Hidden size | 1024 | 768 | 1024 |
| FFN inner hidden size | 4096 | 3072 | 4096 |
| FFN dropout | 0.1 | 0.1 | 0.1 |
| Attention heads | 8 | 12 | 16 |
| Attention dropout | 0.1 | 0.1 | 0.1 |
| Embedding size | 1024 | 1024 | 1024 |
| Stopping criterion | validation patience=15 | training step=150k | training step=150k |
| Batch size | 512 | 2048 | 2048 |
| Learning rate | 2.00E-05 | 1.00E-04 | 1.00E-04 |
| Learning rate schedule | Linear | Linear | Linear |
| Adam $\beta_1$ | 0.9 | 0.98 | 0.98 |
| Adam $\beta_2$ | 0.999 | 0.999 | 0.999 |
| Weight decay | 0 | 0.01 | 0.01 |
| Warmup steps | 4000 | 4000 | 4000 |

Table 7: Hyperparameters used for pre-training.

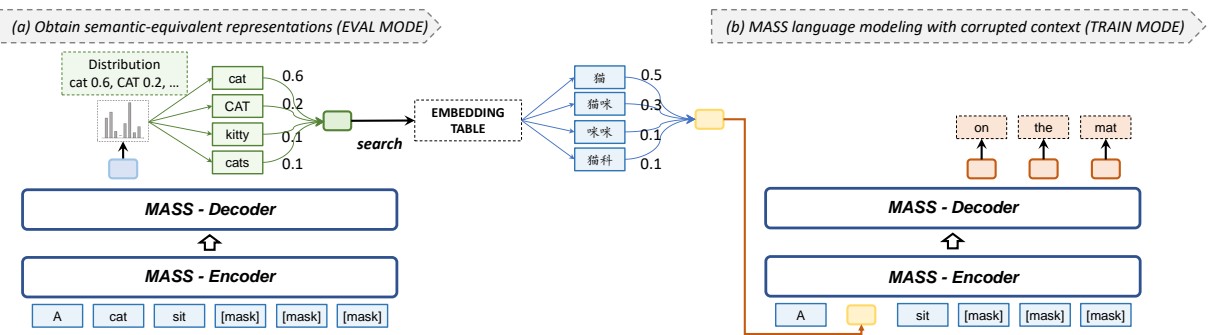

Figure 8: The illustration of our method on MASS language modeling, which masks multiple adjacent tokens for prediction.

| Data | Lan. | # Sent. | Source |
|---|---|---|---|
| En-De | En | 50.0M | (Song et al., 2019) |
| | De | 50.0M | |
| En-Fr/Ro | En | 179.9M | News Crawl 07-17 |
| | Fr | 65.4M | News Crawl 07-17 |
| | Ro | 2.8M | News Crawl 07-17 + WMT16 |

Table 8: Data statistics for unsupervised machine translation training.

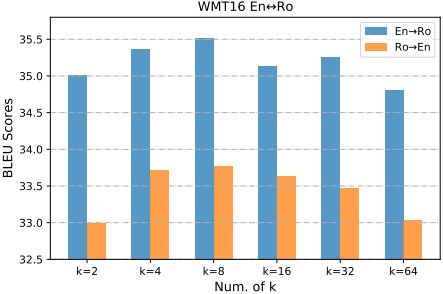

Figure 9: Effect of $k$ on WMT16 En↔Ro.

of pre-trained models, the alignment accuracy increases with the size of the parameters. Moreover, all the models show a similar pattern that distant language pairs have higher alignment accuracy.

Furthermore, we also draw the RSIM scores across layers of different mono-mPLMs in Figure 10. We find that different models share a similar phenomenon that RSIM scores are higher at the lower layers but lower at higher layers. None of them could keep the geometric similarity balanced like para-mPLMs, VECO, or InfoXLM, as shown

in Figure 4. Therefore, we argue that explicit cross-lingual interactions still matter regardless of different architectures.

## D    Ablation Study - Effect of $k$

In the proposed method, SCS-MLM, $k$ plays an important role. Considering that the pre-training of BASE and LARGE models are time-consuming, we pre-train SMALL models with different $k$ and

| Models | params. | #lg | ar | bg | de | el | fr | hi | id | ja | ko | ru | Avg. |
|---|---|---|---|---|---|---|---|---|---|---|---|---|---|
| XLM-RoBERTa | | | | | | | | | | | | | |
| num.of.pairs | | | 1505 | 3322 | 2187 | 1223 | 2164 | 1159 | 2381 | 2501 | 1278 | 4077 | |
| XLM-R-BASE | 250M | 100 | 44.12 | 58.61 | 48.33 | 52.58 | 49.03 | 48.92 | 47.00 | 62.38 | 63.77 | 59.43 | 52.73 |
| XLM-R-LARGE | 560M | 100 | 46.71 | 58.79 | 48.61 | 52.25 | 48.94 | 50.13 | 47.08 | 63.97 | 62.52 | 59.97 | 53.09 |
| Multilingual-T5 | | | | | | | | | | | | | |
| num.of.pairs | | | 817 | 2195 | 2289 | 878 | 1850 | 477 | 1563 | 2176 | 925 | 2631 | |
| mT5-SMALL | 170M | 100 | 42.96 | 59.27 | 53.56 | 48.63 | 55.08 | 41.09 | 49.14 | 55.72 | 61.41 | 62.35 | 52.63 |
| mT5-BASE | 390M | 100 | 53.24 | 64.33 | 57.14 | 58.66 | 62.27 | 52.62 | 56.69 | 65.61 | 71.03 | 69.07 | 60.57 |
| mT5-LARGE | 970M | 100 | 54.96 | 65.51 | 58.41 | 59.34 | 63.57 | 51.36 | 57.90 | 67.63 | 73.41 | 69.49 | 61.75 |
| mT5-XL | 3.2B | 100 | 58.38 | 66.74 | 61.47 | 60.93 | 65.68 | 55.56 | 59.95 | 69.24 | 73.84 | 70.40 | 63.59 |
| X-GLM | | | | | | | | | | | | | |
| num.of.pairs | | | 1271 | 3332 | 3049 | 2212 | 3165 | 1451 | 2862 | 1312 | 1638 | 3636 | |
| XGLM-564M | 560M | 30 | 64.59 | 72.09 | 65.27 | 68.67 | 73.46 | 72.02 | 65.62 | 74.62 | 73.69 | 72.28 | 69.20 |
| XGLM-1.7B | 1.7B | 30 | 64.44 | 71.94 | 64.55 | 69.44 | 73.24 | 71.95 | 65.34 | 76.14 | 73.69 | 73.65 | 69.32 |
| XGLM-2.9B | 2.9B | 30 | 64.99 | 72.36 | 64.81 | 69.62 | 73.74 | 72.50 | 65.62 | 76.14 | 74.05 | 73.98 | 69.67 |
| XGLM-7.5B | 7.5B | 30 | 64.83 | 72.63 | 65.40 | 70.71 | 74.66 | 73.26 | 66.46 | 75.69 | 73.93 | 73.79 | 70.05 |

Table 9: Prediction accuracy of translation pairs derived from different mPLMs across different languages to English. Because XLM-R/mT5/X-GLM supports different numbers of languages and has different vocabularies, comparisons between different types of mPLMs may not be meaningful.

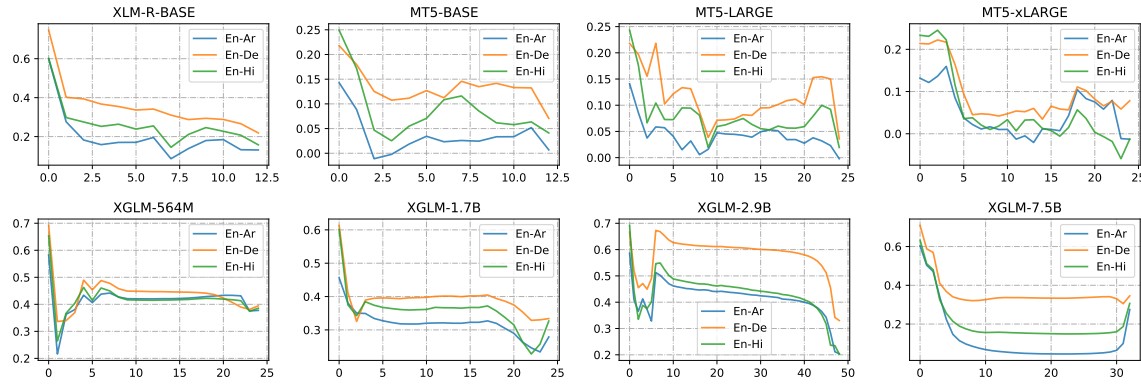

Figure 10: RSIM scores across different layers of different mono-mPLMs.

evaluate the performance on UNMT task (WMT16 En↔Ro). As shown in Figure 9, we can find that SCS-MLM obtains the best performance when $k$ is set as 8. Therefore, we set $k$=8 for all the experiments in our paper.

# E  Experimental Results Details

Due to space limitations, we just report the average cross-lingual transfer metric scores in the main paper. The details for each language test set are listed in Table 10-14.

***Translate-train-all*** is another evaluation method for multilingual pre-trained language models. It means fine-tuning a multilingual model on the concatenation of all data (English training corpus and translated training corpus in other languages). Although our method mainly focuses on improving cross-lingual transfer capability, it can also bring improvements in ***Translate-train-all*** settings.

These results are also provided in Table 10-11.

| Model | de | en | es | fr | ja | ko | zh | avg. |
|---|---|---|---|---|---|---|---|---|
| *Cross-lingual Transfer* | | | | | | | | |
| XLM-R$_{BASE}$ | 86.4 | 93.9 | 88.8 | 88.8 | 76.8 | 76.4 | 79.7 | 84.4 |
| + TCS-MLM | **87.9** | 94.4 | 89.4 | **89.2** | 76.3 | 76.8 | 80.5 | 84.9 |
| + SCS-MLM | **87.9** | **94.5** | 89.0 | 89.0 | **77.5** | **77.0** | **81.0** | **85.1** |
| XLM-R$_{LARGE}$ | 89.7 | 94.7 | 90.1 | 90.4 | 78.7 | 79.0 | 82.3 | 86.4 |
| + TCS-MLM | 90.8 | **95.8** | **91.6** | 91.4 | 81.8 | 81.7 | 84.7 | 88.2 |
| + SCS-MLM | **91.7** | 95.7 | **91.6** | **92.0** | **82.8** | **82.9** | **85.3** | **88.9** |
| *Translate-train-all* | | | | | | | | |
| XLM-R$_{LARGE}$ | **92.2** | **95.7** | 92.7 | 92.5 | 84.7 | 85.9 | 87.1 | 90.1 |
| + SCS-MLM | 91.8 | 95.3 | **93.0** | **93.3** | **86.2** | **87.6** | **88.6** | **90.8** |

Table 10: Cross-lingual transfer results on PAWS-X cross-lingual paraphrase identification for 7 languages.

| Language Pairs | Exported Translation Pairs |
|---|---|
| Bg-En | (_стратегия, _strategi), (_напълно, _completely), (_деца, _children), (_използва, _utiliza), (_постепенно, _gradual) |
| De-En | (_Produkt, _Product), (_Artikel, _Article), (_respect, _respect), (_kultur, _culture), (_Kontakt, _Contact) |
| El-En | (_πρόβλημα, _problem), (_εντελώς, _completely), (_σχέση, _relationship), (_προφανώς, _obviously), (_αποτέλεσμα, _outcome) |
| Fr-En | (_développement, _development), (_present, _present), (_familia, _family), (_pourquoi, _why), (_système, _system) |
| Hi-En | (_रणनीति, _strategi), (_स्थिति, _situation), (_अलग, _separate), (_किलोमीटर, _kilometer), (_रिकॉर्ड, _record) |
| Ja-En | (スポーツ, _sports), (エネルギー, _energy), (投資, _investment), (ビジネス, _business), (システム, _system) |
| Ko-En | (_매우, _extremely), (_프로그램, _programs), (_경제, _economy), (_항상, _always), (_다양한, _various) |
| Mr-En | (_शरिक्कुं, _actually), (_प्रश्नं, _problem), (_अडिस्थान, _basis), (_यात्रा, _voyage), (_कुटुंब, _family) |
| Pt-En | (_milhões, _millions), (_alguém, _someone), (_desenvolvimento, _development), (_praticamentev, _basically), (_técnicas, _techniques) |
| Ru-En | (_модел, _model), (_стратегии, _strategy), (_секс, _sex), (_маркетинг, _marketing), (_персонал, _personal) |
| Si-En | (_වර්ථා, _report), (_චිත්‍ර, _movies), (_සම්පූර්ණ, _complete), (_දැනගත්, _knows), (_ඉතාම, _extremely) |
| Ta-En | (_முக்கிய, _important), (_பயன்படுத்த, _utilize), (_பிரச்சனை, _problem), (_அறிவியல், _science), (_பாதுகாப்பு, _protection) |
| Te-En | (_టెక్నాలజీ, _Technology), (_పూర్తిగా, _completely), (_లైఫ్, _LIFE), (_విడుదల, _release), (_వివిధ, _various) |
| Th-En | (วัฒนธรรม, _cultural), (เกิดขึ้น, _happened), (เทคนิค, _technique), (เทคโนโลยี, _technologies), (ส่วนใหญ่, _mostly) |
| Vi-En | (_triển, _develop), (_sớm, _early), (_giường, _bed), (_thách, _challenge), (_khuyên, _recommend) |
| Zh-En | (工具, _tools), (限制, _limit), (完全, _completely), (改善, _improve), (最新, _latest) |

Figure 11: Some examples that are randomly selected from the exported dictionaries.

| Model | ar | bg | de | el | en | es | fr | hi | ru | sw | th | tr | ur | vi | zh | avg |
|---|---|---|---|---|---|---|---|---|---|---|---|---|---|---|---|---|
| | | | | | | *Cross-Lingual Transfer* | | | | | | | | | | |
| XLM-RBASE | 73.1 | 78.4 | 77.1 | 76.3 | **85.1** | 79.4 | 78.4 | 70.9 | 76.5 | 64.5 | 73.7 | 73.1 | 67.2 | 75.4 | 74.7 | 74.9 |
| + TCS-MLM | 72.7 | 78.2 | 77.3 | 77.0 | 84.1 | **80.3** | **78.8** | 71.4 | 76.4 | 66.9 | 73.0 | 74.2 | 67.6 | 75.4 | 74.4 | 75.2 |
| + SCS-MLM | **73.7** | **79.2** | **78.3** | **77.1** | 84.3 | 79.7 | 78.4 | **72.2** | **77.0** | 67.5 | 73.9 | 75.0 | 68.1 | **76.4** | 75.5 | 75.8 |
| XLM-RLARGE | 77.2 | 83.0 | 82.5 | 80.8 | 88.7 | 83.7 | 82.2 | 75.6 | 79.1 | 71.2 | 77.4 | 78.0 | 71.7 | 79.3 | 78.2 | 79.2 |
| + TCS-MLM | 79.3 | 83.9 | 83.3 | 83.0 | **88.9** | **85.1** | **84.3** | 77.4 | **81.4** | 73.6 | 78.3 | **80.6** | 73.2 | **80.7** | 79.6 | 80.8 |
| + SCS-MLM | **79.9** | **84.1** | **83.5** | 82.9 | 88.8 | 84.8 | **84.3** | 77.7 | 81.3 | 74.2 | 79.1 | 80.6 | 74.4 | 80.7 | 80.2 | 81.1 |
| | | | | | | *Translate-train-all* | | | | | | | | | | |
| XLM-RLARGE | 82.4 | 85.3 | 84.8 | 85.0 | 88.9 | 86.2 | 84.7 | 80.2 | 82.4 | 77.3 | 80.9 | 82.7 | 77.2 | 82.8 | 83.0 | 82.9 |
| + SCS-MLM | 82.9 | 85.4 | 85.2 | 84.9 | 89.0 | 86.0 | 85.0 | 80.8 | 83.0 | 77.8 | 80.9 | 83.1 | 77.8 | 83.0 | 82.6 | 83.2 |

Table 11: Cross-lingual transfer results on XNLI cross-lingual natural language inference for 15 languages.

| | af | ar | bg | de | el | en | es | et | eu | fa | fi | fr | he | hi | hu | id | it |
|---|---|---|---|---|---|---|---|---|---|---|---|---|---|---|---|---|---|
| XLM-RBASE | 88.2 | 67.1 | 88.7 | 88.6 | 86.2 | 95.9 | 88.4 | 86.7 | 67.7 | 69.7 | 86.3 | 86.7 | 67.9 | 73.2 | 83.0 | 72.8 | 88.9 |
| + TCS-MLM | 88.3 | 66.8 | 88.8 | 88.6 | 85.8 | 95.9 | 88.3 | 85.8 | 68.3 | 69.6 | 85.3 | 86.6 | 67.1 | 71.9 | 82.9 | 72.5 | 88.9 |
| + SCS-MLM | 89.0 | 65.7 | 88.8 | 89.6 | 87.5 | 95.7 | 87.4 | 85.1 | 71.6 | 69.1 | 85.0 | 86.3 | 66.3 | 67.5 | 82.4 | 72.4 | 88.6 |
| XLM-RLARGE | 89.8 | 67.5 | 88.1 | 88.5 | 86.3 | 96.1 | 88.3 | 86.5 | 72.5 | 70.6 | 85.8 | 87.2 | 68.3 | 76.4 | 82.6 | 72.4 | 89.4 |
| + TCS-MLM | 89.4 | 67.2 | 87.0 | 88.8 | 87.4 | 96.1 | 88.2 | 86.3 | 76.9 | 71.2 | 86.4 | 86.3 | 69.0 | 72.7 | 83.1 | 73.0 | 90.1 |
| + SCS-MLM | 89.4 | 69.1 | 88.7 | 88.6 | 86.6 | 96.1 | 87.7 | 86.5 | 71.6 | 71.0 | 85.8 | 87.0 | 68.2 | 74.3 | 83.2 | 72.7 | 88.5 |

| | ja | kk | ko | mr | nl | pt | ru | ta | te | th | tl | tr | ur | vi | yo | zh | avg. |
|---|---|---|---|---|---|---|---|---|---|---|---|---|---|---|---|---|---|
| XLM-RBASE | 27.9 | 76.3 | 53.3 | 82.6 | 89.3 | 88.0 | 89.3 | 62.7 | 84.8 | 45.2 | 89.9 | 74.5 | 64.1 | 57.9 | 24.3 | 33.4 | 73.6 |
| + TCS-MLM | 23.1 | 76.9 | 52.6 | 81.7 | 89.3 | 88.1 | 89.3 | 61.8 | 85.6 | 43.0 | 91.9 | 74.2 | 62.1 | 57.5 | 25.9 | 29.9 | 73.2 |
| + SCS-MLM | 31.5 | 77.1 | 53.8 | 79.3 | 89.5 | 86.7 | 89.2 | 63.4 | 84.8 | 49.9 | 91.2 | 73.9 | 58.9 | 55.2 | 27.8 | 44.8 | 73.8 |
| XLM-RLARGE | 15.9 | 78.1 | 53.9 | 80.8 | 89.5 | 87.6 | 89.5 | 65.2 | 86.6 | 47.2 | 92.2 | 76.3 | 70.3 | 56.8 | 24.6 | 25.7 | 73.8 |
| + TCS-MLM | 33.5 | 79.0 | 52.9 | 85.7 | 89.3 | 87.2 | 89.6 | 62.8 | 85.0 | 46.7 | 93.5 | 76.3 | 67.8 | 60.0 | 33.9 | 35.7 | 75.1 |
| + SCS-MLM | 35.2 | 78.4 | 53.2 | 83.9 | 89.6 | 87.6 | 89.9 | 62.8 | 86.3 | 54.4 | 93.9 | 75.4 | 68.8 | 58.0 | 39.5 | 54.9 | 76.0 |

Table 12: Cross-lingual transfer results on UDPOS (POS) cross-lingual pos-tagging for 33 languages.

| Model | af | ar | bg | bn | de | el | en | es | et | eu | fa | fi | fr | he | hi | hu | id | it | ja | jv |
|---|---|---|---|---|---|---|---|---|---|---|---|---|---|---|---|---|---|---|---|---|
| XLM-R$_{\text{BASE}}$ | 76.2 | 55.5 | 78.6 | 71.8 | 75.0 | 75.8 | 81.8 | 69.2 | 71.7 | 58.7 | 54.5 | 76.2 | 76.2 | 53.0 | 69.3 | 77.7 | 48.5 | 78.6 | 17.7 | 58.7 |
| + TCS-MLM | 78.1 | 52.9 | 78.8 | 73.2 | 75.5 | 75.8 | 81.9 | 73.4 | 72.2 | 58.1 | 54.4 | 76.3 | 77.3 | 52.8 | 69.4 | 77.5 | 50.7 | 78.0 | 18.8 | 59.5 |
| + SCS-MLM | 77.8 | 49.9 | 79.9 | 70.0 | 75.1 | 76.0 | 81.5 | 76.7 | 72.5 | 61.0 | 50.1 | 76.5 | 77.6 | 55.2 | 69.3 | 77.6 | 48.9 | 78.1 | 21.3 | 63.7 |
| XLM-R$_{\text{LARGE}}$ | 78.9 | 53.0 | 81.4 | 78.8 | 78.8 | 79.5 | 84.7 | 79.6 | 79.1 | 60.9 | 61.9 | 79.2 | 80.5 | 56.8 | 73.0 | 79.8 | 53.0 | 81.3 | 23.2 | 62.5 |
| + TCS-MLM | 79.4 | 47.8 | 82.0 | 75.3 | 79.9 | 81.0 | 83.4 | 72.1 | 81.6 | 72.3 | 55.7 | 81.4 | 81.3 | 55.0 | 72.7 | 81.8 | 65.3 | 82.4 | 19.3 | 64.3 |
| + SCS-MLM | 78.8 | 55.1 | 83.0 | 76.8 | 79.9 | 79.3 | 83.7 | 77.0 | 80.8 | 68.0 | 63.7 | 81.1 | 79.7 | 55.3 | 72.4 | 81.2 | 56.4 | 81.8 | 17.6 | 64.9 |
| | ka | kk | ko | ml | mr | ms | my | nl | pt | ru | sw | ta | te | th | tl | tr | ur | vi | yo | zh |
| XLM-R$_{\text{BASE}}$ | 67.5 | 43.4 | 52.0 | 61.8 | 62.3 | 59.9 | 51.2 | 80.1 | 78.7 | 64.9 | 69.3 | 55.1 | 49.6 | 0.0 | 71.6 | 75.2 | 66.0 | 67.7 | 49.5 | 25.8 |
| + TCS-MLM | 68.3 | 43.3 | 54.7 | 63.1 | 63.1 | 58.0 | 54.5 | 81.0 | 77.7 | 64.8 | 68.2 | 58.5 | 53.3 | 0.0 | 72.4 | 77.7 | 68.5 | 68.9 | 50.0 | 26.4 |
| + SCS-MLM | 69.4 | 40.8 | 52.7 | 65.0 | 63.1 | 53.4 | 56.1 | 80.5 | 78.9 | 64.6 | 69.4 | 59.4 | 55.1 | 0.0 | 70.8 | 79.4 | 67.7 | 68.6 | 43.1 | 28.5 |
| XLM-R$_{\text{LARGE}}$ | 71.6 | 56.2 | 60.0 | 67.8 | 68.1 | 57.1 | 54.3 | 84.0 | 81.9 | 69.1 | 70.5 | 59.5 | 55.8 | 1.3 | 73.2 | 76.1 | 56.4 | 79.4 | 33.6 | 33.1 |
| + TCS-MLM | 68.4 | 55.2 | 60.7 | 62.2 | 67.4 | 66.7 | 55.6 | 84.9 | 82.9 | 69.7 | 69.8 | 59.0 | 55.5 | 0.0 | 74.3 | 84.1 | 71.6 | 74.7 | 36.9 | 27.5 |
| + SCS-MLM | 68.1 | 53.9 | 60.4 | 63.3 | 65.0 | 69.1 | 59.0 | 84.1 | 83.1 | 71.8 | 69.5 | 59.7 | 55.8 | 0.0 | 74.5 | 82.5 | 75.0 | 78.6 | 47.7 | 26.1 |

Table 13: Cross-lingual transfer results on Wikiann (NER) cross-lingual name entity recognition for 40 languages.

| | af | ar | bg | bn | de | el | es | et | eu | fa | fi | fr | he | hi | hu | id | it | ja |
|---|---|---|---|---|---|---|---|---|---|---|---|---|---|---|---|---|---|---|
| XLM-R$_{\text{BASE}}$ | 52.0 | 35.9 | 67.1 | 29.3 | 88.9 | 53.7 | 70.4 | 43.7 | 24.3 | 64.8 | 64.0 | 73.2 | 50.8 | 50.4 | 58.5 | 68.8 | 65.2 | 53.8 |
| + TCS-MLM | 62.6 | 42.6 | **73.7** | **36.4** | **92.3** | **60.1** | 78.6 | 57.6 | 40.4 | **71.8** | 72.6 | **79.9** | 56.2 | **65.9** | 64.7 | 76.7 | 69.3 | 50.4 |
| + SCS-MLM | **64.2** | **42.7** | 73.3 | 33.5 | 92.0 | 59.6 | **78.9** | **59.4** | **41.3** | 70.3 | **73.5** | 79.1 | **57.1** | 62.8 | **65.1** | **77.3** | **70.0** | 50.9 |
| | jv | ka | kk | ko | ml | mr | nl | pt | ru | sw | ta | te | th | tl | tr | ur | vi | zh |
| XLM-R$_{\text{BASE}}$ | 15.1 | 36.2 | 32.7 | 49.3 | 54.6 | 37.4 | 77.2 | 77.6 | 70.2 | 15.9 | 24.1 | 28.6 | 32.3 | 29.0 | 55.2 | 27.7 | 67.7 | 59.7 |
| + TCS-MLM | 17.1 | 42.6 | 44.9 | 54.6 | 56.3 | 45.3 | **83.0** | 82.9 | **76.1** | **23.6** | 26.7 | **30.8** | **42.9** | 37.5 | 61.1 | **41.1** | 73.6 | 60.3 |
| + SCS-MLM | **18.0** | **43.2** | **45.2** | **55.7** | **57.6** | **46.9** | 82.6 | **83.1** | 75.2 | 22.8 | **27.0** | 28.6 | 41.6 | **37.8** | **63.5** | 40.9 | **74.6** | **60.9** |

Table 14: Cross-lingual transfer results on Tatoeba cross-lingual sentence retrieval for 34 languages.