# OpenReview forum: "Take a Closer Look at Multilinguality! Improve Multilingual Pre-Training Using Monolingual Corpora Only"
_EMNLP/2023/Conference — EMNLP 2023 Findings_

### Official Review · Reviewer_R5qn · 2023-08-03

**Soundness:** 3

**Excitement:**

3: Ambivalent: It has merits (e.g., it reports state-of-the-art results, the idea is nice), but there are key weaknesses (e.g., it describes incremental work), and it can significantly benefit from another round of revision. However, I won't object to accepting it if my co-reviewers champion it.

**Missing References:**

- Reid and Artetxe. PARADISE: Exploiting Parallel Data for Multilingual Sequence-to-Sequence Pretraining. NAACL 2022
- Ding et al. A Simple and Effective Method to Improve Zero-Shot Cross-Lingual
Transfer Learning. COLING 2022

**Paper Topic And Main Contributions:**

Current multilingual pretrained language models (mPLMs) are typically pretrained with monolingual corpora for multiple languages, i.e., mono-mPLMs, and may be also further improved using parallel resources, i.e., para-mPLMs. Due to the limited available parallel resource, this paper aims to improve mPLMs without parallel resources.

The paper firstly conduct analyses to investigate the token properties of a mPLM, i.e., XLM-R, and the results show that cross-lingual token alignments (geometrically aligned instead of absolutely aligned) occur at the embedding layer but become weaker at the higher layers.

Based on these observations, this paper proposes an training objective of token-level and semantic-level code-switched masked language modeling, which employs the self-induced token alignments to explicitly improve cross-lingual interactions over layers of mono-mPLMs without relying on parallel sentences.

The experimental results show that the proposed method outperforms mono-mPLMs and achieve comparable results with para-mPLMs on various natural language understanding tasks.

**Questions For The Authors:**

A.Line 225 and Figure 3(a), average cos between translation pairs. For computing the sentence embeddings, is the method averaging token embeddings of the input seqence? Why are there different results for token-level and sentence-level semantic alignment?

B.Line 310-311 and Figure 5, how does the output distribution computed? For the input, the token ‘cat’ is provided instead of masked. Does it make sense?

C.Line 352, why is $\alpha$ set as 0.7, which is different from XLM-R?

D.Is it necessary to use self-induced lexicons instead of gold lexicons? For high-resource languages, the gold lexicons are usually easy to obtain. For low-resource languages, the self-induced lexicons may be in low-quality, leading to worse results on downstream tasks.

**Reasons To Accept:**

- This paper proposes new training objectives to improve mono-mPLMs using self-induced token alignments instead of external parallel resources.
- The experimental results show that it can outperform mono-mPLMs and achieve comparable results with para-mPLMs

**Reasons To Reject:**

The contributions are incremental. Specifically,
1. Token alignment analysis (Cao et al. Multilingual Alignment of Contextual Word Representations. ICLR 2020.)
2. Token-level code-switched masked language modeling (Reid and Artetxe. PARADISE: Exploiting Parallel Data for Multilingual Sequence-to-Sequence Pretraining. NAACL 2022)

**Reproducibility:**

3: Could reproduce the results with some difficulty. The settings of parameters are underspecified or subjectively determined; the training/evaluation data are not widely available.

**Reviewer Confidence:**

5: Positive that my evaluation is correct. I read the paper very carefully and I am very familiar with related work.

---

> ### Author Rebuttal · Authors · 2023-08-28
>
> We appreciate your thoughtful insights and comments on our work.
>
> **About the Difference with Previous Studies**
>
> We would like to clarify the distinctions between our work and the notable contributions of [1] and [2] in multilingual pre-training. The main difference is utilizing labeled bilingual resources to improve multilinguality [1-2] or exploiting the self-improving approach without reliance on external parallel data.
>
> * *Regarding Scenarios and Motivation*:
>   * [1] and [2] focus on enhancing cross-lingual capabilities by utilizing human-labeled parallel resources, such as bilingual sentences with token alignments[1] or gold dictionaries[2]. They are suitable for scenarios when parallel resources are available.
>   * By contrast, our work unveils the presence of cross-lingual token alignments within mPLMs pre-trained exclusively on monolingual corpora. Moreover, we further demonstrate the self-improving capability of mono-mPLMs through the incorporation of self-induced token alignments.
> * *Regarding Methodology:*
>   * [1] uses L2 loss to draw token representations closer with the help of labeled word alignments, and [2] leverages gold dictionaries from MUSE and Flores to conduct monolingual and parallel denoising tasks, requiring the model to recover the original tokens with substituted translations.
>   * In contrast, our approach utilizes self-induced token alignments to construct context, forcing the model to predict *&lt;mask&gt;* tokens with the semantic-equivalent but code-switched surrounding tokens and thus improving the multilinguality.
>   * Moreover, [1] primarily focuses on the token alignments within a specific parallel sentence pair, while our analysis pertains to the comparison of two language-specific vocabularies.
>
> **About the Questions**
>
> * Q.A - About Average Cosine of Translation Pairs.
>   * We apologize for the misleading regarding "translation pairs". In this paper, the "translation pairs" are token-level instead of sentence-level. For example, "happy - سعيد" and "interesting - दिलचस्प". The average cosine means the average scores of 500 token translation pairs for a particular language direction (dash lines in Figure 3(a)). We can find that the average cosine scores become larger, demonstrating that distinct language token representations aggregate at higher layers. By contrast, the token alignments become weaker (solid lines in Figure 3(a)).
>   * The sentence-level alignments (using average pooling of token representations as the sentence representation) are regarded to appear in the middle layer of mPLMs [3-4]. We think that lower layers focus on the token-level features while middle layers focus on the syntactic and sentence-level features [5-6], which result in the difference between token-level and sentence-level alignments.
>
> * Q.B - About the Computation of Distribution.
>   * The distribution $d$ is calculated as $d = hW$, where $h$ denotes hidden states and $W$ is the output projection weights, which are usually tied with the embedding weights.
>   * We believe that replacing context tokens instead of masked positions is meaningful. Masked language modeling (MLM) supposes that the masked tokens can be predicted by the understanding of context. When the context tokens are replaced by the semantic-equivalent tokens in other languages, the model should pay attention to the replaced tokens and perform the correct prediction.
>
>     For the sentence "*We योजना to mennä to the &lt;mask&gt; to see 熊猫* ", we record the attention weights from *&lt;mask&gt;* to other tokens of XLM-R and our SCS-MLM in Table 1. We find that SCS-MLM pays more attention to "*see*", "熊" and "猫", achieving a higher prediction probability of the gold token "_zoo" (0.2862) than XLM-R (0.1046).
>   * Table 2 also shows that SCS-MLM validation loss also gradually decreases during training, corroborating the effectiveness of our approach.
>
>
>
>   **Table 1. Attention weights from *&lt;mask&gt;* to other tokens of XLM-R and SCS-MLM. We also label the gold output "_zoo" and the prediction probability.**
>
> |                | &lt;s&gt; | ▁We   | ▁योजना | ▁to   | ▁mennä | ▁to    | ▁the   | &lt;mask&gt;    | ▁to    | ▁see      | ▁         | *熊*        | *猫*        | &lt;/s&gt; |
> | -------------- | :-------: | ----- | ------ | ----- | ------ | ------ | ------ | --------------- | ------ | --------- | --------- | --------- | --------- | ---------- |
> | Attn(XLM-R)          |   0.330   | 0.014 | 0.000  | 0.009 | 0.052  | 0.062  | 0.079  | 0.004           | 0.029  | 0.035     | 0.021     | 0.015     | 0.018     | 0.333      |
> | - *Prediction* |           |       |        |       |        |        |        | ▁zoo - p=0.1046 |        |           |           |           |           |            |
> | Attn(SCS-MLM)        |   0.285   | 0.014 | 0.000  | 0.009 | 0.035  | 0.059  | 0.070  | 0.003           | 0.026  | 0.082 | 0.036 | **0.029** | **0.043** | 0.309      |
> | - *Prediction* |           |       |        |       |        |        |        | ▁zoo - p=0.2862 |        |           |           |           |           |            |
> | $\Delta$ = Attn(SCS-MLM) - Attn(XLM-R)      |  -0.045   | 0.000 | 0.000  | 0.000 | -0.017 | -0.003 | -0.009 | -0.001          | -0.003 | 0.047 | 0.015 | **0.014** | **0.025** | -0.024     |
>
>
>
> **Table 2. SCS-MLM loss gradually decreases with the increase of training steps.**
>
> | Training Step           | 1k   | 5k   | 10k  | 20k  | 50k  | 100k | 150k |
> | ----------------------- | ---- | ---- | ---- | ---- | ---- | ---- | ---- |
> | SCS-MLM Validation Loss | 2.35 | 2.30 | 2.25 | 2.21 | 2.17 | 2.14 | 2.12 |
>
>
>
>
> * Q.C - About the Sampling Rate.
>
>   * The sampling rate relies on the data sizes of different languages. The sizes of our pre-training corpora are different from XLM-R, leading to different $\alpha$ settings like previous studies (XLM-R[7]: $\alpha=0.3$, VECO[8]: $\alpha=0.5$, InfoXLM[4]: $\alpha=0.7$). For our data, we find that setting $\alpha=0.7$ would lead to balanced sampling probabilities for different languages.
>
> * Q.D - The Necessity of Self-Induced Dictionary.
>
>
>   * First, our work introduces valuable insights regarding the potential of exporting cross-lingual dictionaries from mono-mPLMs (multilingual language models pre-trained with multiple monolingual corpora). The proposed method further demonstrates that self-induced dictionaries can enhance the performance of mPLMs like conventional gold dictionaries.
>
>   * Second, we classify the availability of cross-lingual dictionaries based on the resources involved, as shown in Table 3.
>
>     * **CLASS-A:** This category usually has high-resource monolingual corpora and parallel corpora, like widely-used English-centric pairs such as En-Fr and En-Zh. Gold dictionaries for these pairs are readily accessible either as open-source resources ([MUSE](https://github.com/facebookresearch/MUSE), [wikidict](https://github.com/open-dict-data/wikidict-en)) or through extraction from parallel sentences. Our work demonstrates that mPLMs also support producing dictionaries for these language directions with good quality.
>     * **CLASS-B:**  Involving high-resource monolingual corpora but limited parallel corpora, this class encapsulates most language pairs worldwide. In this scenario, acquiring gold dictionaries becomes a challenge. For instance, language pairs like Greek-Hindi (El-Hi) and Tamil-Japanese (Ta-Ja) boast abundant monolingual sentences but lack open-source gold dictionaries. In contrast, our work underlines that mono-mPLMs can effectively yield reliable token translations (self-induced dictionaries), emerging as a viable alternative under these circumstances.
>       We randomly choose some exported token translation pairs of El-Hi and Ta-Ja, which are presented in Table 4-5.
>     * **CLASS-C:** This category involves both low-resource monolingual corpora and parallel corpora, making the acquisition of gold dictionaries, as well as the generation of dictionaries from mPLMs challenging.  Finding effective solutions for these languages necessitates other methodologies and resources. Nonetheless, this class pertains to just a minute fraction of languages. To illustrate, among the 50 languages that we processed, only yo, gd, and fy have this problem.
>
>     In practical scenarios, the availability of gold dictionaries dictates the direct use of such resources. However, in their absence, leveraging self-induced dictionaries from mPLMs emerges as a pragmatic strategy to augment the cross-lingual capabilities of models.
>
>
>
>   **Table 3. The availability of cross-lingual dictionaries for different resource scenarios.**
>
>   |  Type   |  Monolingual  |   Parallel    | Gold Dictionary | Self-induced Dictionary |
>   | :-----: | :-----------: | :-----------: | :-------------: | :---------------------: |
>   | CLASS-A | High-resource | High-resource |     &check;     |         &check;         |
>   | CLASS-B | High-resource | Low-resource  |     &cross;     |         &check;         |
>   | CLASS-C | Low-resource  | Low-resource  |     &cross;     |         &cross;         |
>
>   **Table 4. Some randomly selected Ja-Ta token translations derived from our XLM-R.**
>
>   | Japanese Token | Tamil Token | English Meaning |
>   | -------------- | ----------- | ---------------- |
>   | それを         | ▁அதை        | it               |
>   | 銀行           | ▁வங்கி       | bank             |
>   | 生まれた       | ▁பிறந்த      | born             |
>   | なぜ           | ▁ஏன்         | why              |
>   | 重要な         | ▁முக்கியமான  | important        |
>   | ということを   | ▁என்பதை      | that             |
>   | 理由は         | ▁காரணம்      | the reason       |
>   | 市場           | ▁சந்தை       | market           |
>   | 今後の         | ▁எதிர்கால    | future           |
>   | 統治           | ▁ஆட்சி       | reign / rule     |
>   | 様々な         | ▁பல்வேறு     | various          |
>   | 最終           | ▁இறுதி      | final            |
>
>   **Table 5. Some randomly selected El-Hi token translations derived from XLM-R.**
>
>   | Greek Token | Hindi Token | English Meaning |
>   | ----------- | ----------- | ---------------- |
>   | ▁ξαφνικά    | ▁अचानक      | suddenly         |
>   | ▁παράδειγμα | ▁उदाहरण     | example          |
>   | ▁φύση       | ▁प्रकृति      | nature           |
>   | ▁ποιότητας  | ▁गुणवत्ता     | quality          |
>   | ▁τοπική     | ▁स्थानीय     | local            |
>   | ▁επίσημη    | ▁आधिकारिक   | official         |
>   | ▁ενδιαφέρον | ▁दिलचस्प     | interesting      |
>   | ▁τύπου      | ▁टाइप       | type             |
>   | ▁βίντεο     | ▁वीडियो     | video            |
>   | ▁Προσωπικ   | ▁व्यक्तिगत    | personal         |
>   | ▁τραγούδι   | ▁गाने        | songs            |
>   | ▁δεκαετίες  | ▁दशक        | decade           |
>
>
>
>   **About the Missing References**
>
>   We will include the missing references in the revised paper.
>
>
>
>   Thank you once more for your comments. We think the discussion about these issues would improve our paper.
>
>
>
>
>   **References**
>
>   [1] Steven Cao, Nikita Kitaev, and Dan Klein. 2020. Multilingual Alignment of Contextual Word Representations. In Proc. of ICLR 2020.
>
>   [2] Machel Reid, Mikel Artetxe. 2022. PARADISE: Exploiting Parallel Data for Multilingual Sequence-to-Sequence Pretraining. In Proc. of NAACL 2022.
>
>   [3] Junjie Hu, Sebastian Ruder, Aditya Siddhant, Graham Neubig, Orhan Firat, and Melvin Johnson. 2020. XTREME: A massively multilingual multi task benchmark for evaluating cross-lingual generalisation. In Proc. of ICML 2020.
>
>   [4] Zewen Chi, Li Dong, Furu Wei, Nan Yang, Saksham Singhal, Wenhui Wang, Xia Song, Xian-Ling Mao, Heyan Huang, and Ming Zhou. 2021. InfoXLM: An information-theoretic framework for cross-lingual language model pre-training. In Proc. of NAACL 2021.
>
>   [5] Ganesh Jawahar, Benoît Sagot, and Djamé Seddah. 2019. What Does BERT Learn about the Structure of Language? In Proc. of ACL 2019.
>
>   [6] Shijie Wu and Mark Dredze. 2019. Beto, Bentz, Becas: The Surprising Cross-Lingual Effectiveness of BERT. In Proc. of EMNLP 2019.
>
>   [7] Alexis Conneau, Kartikay Khandelwal, Naman Goyal, Vishrav Chaudhary, Guillaume Wenzek, Francisco Guzmán, Edouard Grave, Myle Ott, Luke Zettlemoyer, and Veselin Stoyanov. 2020. Unsupervised cross-lingual representation learning at scale. In Prof .of ACL 2020.
>
>   [8] Fuli Luo, Wei Wang, Jiahao Liu, Yijia Liu, Bin Bi, Song fang Huang, Fei Huang, and Luo Si. 2021. VECO: Variable and flexible cross-lingual pre-training for language understanding and generation. In Proc. of ACL-IJCNLP 2021.

---

### Official Review · Reviewer_Swjy · 2023-08-05

**Soundness:** 4

**Excitement:**

3: Ambivalent: It has merits (e.g., it reports state-of-the-art results, the idea is nice), but there are key weaknesses (e.g., it describes incremental work), and it can significantly benefit from another round of revision. However, I won't object to accepting it if my co-reviewers champion it.

**Missing References:**

There is a parallel work from ACL 2023, though not a reason to reject.  The parallel paper discusses a very similar idea (self-induce candidates and weighted aggregation) but focuses on masking strategies instead of data augmentation (this paper).

Xi Ai and Bin Fang. 2023. On-the-fly Cross-lingual Masking for Multilingual Pre-training. In Proceedings of the 61st Annual Meeting of the Association for Computational Linguistics (Volume 1: Long Papers), pages 855–876, Toronto, Canada. Association for Computational Linguistics.

**Paper Topic And Main Contributions:**

This paper compares parallel-corpora-based multilingual training (para-mPLM) with monolingual-corpora-based multilingual training (mono-mPLM) by analyzing the embedding space and the top-layer space. The author found the geometric similarity of representations across different languages contributes to multilingualism but it is not observed in the top-layer space of mono-mPLM. The authors further argue that cross-lingual interactions at the bottom layer can help the model preserve the geometric similarity for the top-layer space.

To this end, the authors present a data augmentation method based on cross-lingual word substitutions. To find suitable cross-lingual word substitutions, the authors use mono-mPLM to predict the distribution of a word and search for corresponding translations in the embedding space. Then, a weighted representation of all the translations is used to replace the original word.

**Questions For The Authors:**

Please refer to Reasons To Reject.

1. Some details are missing. How many tokens do you consider for top-k (eq. 4)?

2. In line 289 ("Therefore, we replace 10∼15% tokens using the dictionary to construct multilingual code-switched"), how do you decide 10%, 15%, or others?

**Reasons To Accept:**

1. The presentation is good.
2. The authors present sufficient and clear empirical studies to justify the motivation.
3. Experiments including 5 tasks support the presented results.

**Reasons To Reject:**

1. My major concern is about the details of experimental setups. Concretely, the authors use mono-mPLM to infer substations. However, mono-mPLM is randomly initialized, which means that mono-mPLM only makes random substitutions in early training. Do the authors use pre-trained mono-mPLM? Is the presented method a post-training method? Do random substitutions in early training not impact the final performance? How many tokens do you use for weighted aggregation?

2.  The authors might justify the effectiveness by showing attention weights on the cross-lingual word substitutions. The model might ignore the cross-lingual word substitutions if the context is sufficient for prediction. For instance, the model just predicts some simple stop words.


I am very happy to discuss these issues.

**Reproducibility:**

4: Could mostly reproduce the results, but there may be some variation because of sample variance or minor variations in their interpretation of the protocol or method.

**Reviewer Confidence:**

5: Positive that my evaluation is correct. I read the paper very carefully and I am very familiar with related work.

---

> ### Author Rebuttal · Authors · 2023-08-28
>
> We express our gratitude for your insightful comments, which will help us improve our paper.
>
> **About the Experimental Details**
>
> * Our model is initialized by pre-trained XLM-R and performs second-stage pre-training, akin to prior works such as VECO[1], HICTL[2], InfoXLM[3] and ERNIE-M[4]. Therefore, the replacements are meaningful during training.
> * We use Top-8 tokens for the weighted aggregation. As we analyzed in Appendix C (Figure 9 is placed in the appendix due to the space limits), we find that setting *k* as 8 would obtain the best performance on the downstream task (UNMT). We will clarify this aspect in the main pages of the revised paper.
> * Regarding the replacement ratio.
>
>   * We take a specific example to elaborate. Suppose we have the dictionary *D* with *n* token translation pairs for language En-X and English has *m* tokens. Then, we define the cover rate as *q=n/m* and the replacement rate is *p=0.15/q*. For token *t* in the current batch, if *t* exists in the dictionary *D*, we use a random function to sample a number 0<=*s*<=1. When *s < p*, we replace token *t* with its translations. However, due to the dynamic changes of tokens in different batches, the actual replacement rates may be slightly varied. Thus, we state that "we replace 10%-15% tokens ..." in our paper.
>   * Furthermore, some of our preliminary experiments have shown that: when the replacement ratio is too high, the model struggles to make accurate predictions due to the heightened levels of language ambiguity. On the contrary, when the replacement ratio is low, the potential improvements can be restricted. A range of 10-15% is a favorable selection.
> * Finally, we will open-source our code, experimental details, pre-trained language models and the self-induced dictionaries.
>
> **About the Attention Weights**
>
> * We use the following example to illustrate the effectiveness of cross-lingual token substitutions. We use the translation pairs (plan - योजना), (go - mennä), (pandas - 熊猫) to replace the following English sentence and predict the *&lt;mask&gt;* (*_zoo*) token:
>
>   *We plan to go to the zoo to see pandas*         &rArr;       *We योजना to mennä to the &lt;mask&gt; to see 熊猫*
>
>   We extract the attention weights from *&lt;mask&gt;* to other tokens in the last layer of XLM-R and our model. We find that:
>
>   * &lt;s&gt; and &lt;/s&gt; have the highest attention weights than other tokens.
>   * The golden token "▁zoo" can be predicted with higher probability by SCS-MLM (0.2862) than XLM-R (0.1046).
>   * The contrast in attention weights for specific tokens (*see*, 熊, 猫) at the *<mask>* position underscores our model's ability to focus more intently on semantically equivalent yet linguistically different tokens compared to vanilla XLM-R. The positive values of $\Delta$ = Attn(SCS-MLM) - Attn(XLM-R) demonstrate the effectiveness of our method.
>
>   We agree with you that for basic stop words, the assistance of cross-lingual context tokens might be unnecessary.
>
>
>
> **Table 1. Attention weights from *&lt;mask&gt;* to other tokens of XLM-R and SCS-MLM. We also label the gold output "_zoo" and the prediction probability.**
>
> |                | &lt;s&gt; | ▁We   | ▁योजना | ▁to   | ▁mennä | ▁to    | ▁the   | &lt;mask&gt;    | ▁to    | ▁see      | ▁         | *熊*        | *猫*        | &lt;/s&gt; |
> | -------------- | :-------: | ----- | ------ | ----- | ------ | ------ | ------ | --------------- | ------ | --------- | --------- | --------- | --------- | ---------- |
> | Attn(XLM-R)          |   0.330   | 0.014 | 0.000  | 0.009 | 0.052  | 0.062  | 0.079  | 0.004           | 0.029  | 0.035     | 0.021     | 0.015     | 0.018     | 0.333      |
> | - *prediction* |           |       |        |       |        |        |        | ▁zoo - p=0.1046 |        |           |           |           |           |            |
> | Attn(SCS-MLM)        |   0.285   | 0.014 | 0.000  | 0.009 | 0.035  | 0.059  | 0.070  | 0.003           | 0.026  | 0.082 | 0.036 | **0.029** | **0.043** | 0.309      |
> | - *prediction* |           |       |        |       |        |        |        | ▁zoo - p=0.2862 |        |           |           |           |           |            |
> | $\Delta$ = Attn(SCS-MLM) - Attn(XLM-R)      |  -0.045   | 0.000 | 0.000  | 0.000 | -0.017 | -0.003 | -0.009 | -0.001          | -0.003 | 0.047 | 0.015 | **0.014** | **0.025** | -0.024     |
>
> **About the Missing Reference**
>
> * Following your suggestion, we read this paper [5]. It is an impressive work that explores the *MASK* strategy to improve cross-lingual pre-training, which uses average weighted embedding as the "prototype-word" to replace *<mask>*, enhancing the cross-lingual knowledge for prediction. Different from [5], our study requests the model to perform correct prediction with semantically equivalent but code-switched contexts, improving the context information usage during prediction. Regrettably, the timeline of EMNLP 2023 submission predates the release of ACL 2023 publications, hence the absence of discussion on this work in our paper. We would like to include this work and discuss the difference between our method and theirs in our revised version.
>
>
>
> Thanks again for your thorough review! We sincerely hope that our response addresses your questions and concerns.
>
>
>
> **References**
>
> [1] Fuli Luo, Wei Wang, Jiahao Liu, Yijia Liu, Bin Bi, Song fang Huang, Fei Huang, and Luo Si. 2021. VECO: Variable and flexible cross-lingual pre-training for language understanding and generation. In Proc. of ACL 2021.
>
> [2] Xiangpeng Wei, Rongxiang Weng, Yue Hu, Luxi Xing, Heng Yu, and Weihua Luo. 2021. On learning univer827 sal representations across languages. In Proc. of ICLR 2021.
>
> [3] Zewen Chi, Li Dong, Furu Wei, Nan Yang, Saksham Singhal, Wenhui Wang, Xia Song, Xian-Ling Mao, Heyan Huang, and Ming Zhou. 2021. InfoXLM: An information-theoretic framework for cross-lingual language model pre-training. In Proc. of NAACL 2021.
>
> [4] Xuan Ouyang, Shuohuan Wang, Chao Pang, Yu Sun, Hao Tian, Hua Wu, and Haifeng Wang. 2021. ERNIE-M: Enhanced multilingual representation by aligning cross-lingual semantics with monolingual corpora. In Proc. of EMNLP 2021.
>
> [5] Xi Ai and Bin Fang. 2023. On-the-fly Cross-lingual Masking for Multilingual Pre-training. In Proc. of ACL 2023.

---

### Official Review · Reviewer_Jsy5 · 2023-08-10

**Paper Topic And Main Contributions:** 1. This paper investigates the alignm…
**Soundness:** 4

**Excitement:**

4: Strong: This paper deepens the understanding of some phenomenon or lowers the barriers to an existing research direction.

**Questions For The Authors:**

1. Would other MPLMs likewise benefit from the approach you propose? It would make the paper more sound if experiments could be conducted in multiple mPLMs.

**Reasons To Accept:**

1. This paper is well-written and well-presented.

2. This paper attempts to improve MPLM using monolingual corpora only and significantly improve the performance compared with the strong baselines.

3. Further analysis demonstrates that the methods improve the geometric similarity of representations for different languages, and thus promoting the cross-lingual transfer capability.

4. The designed empirical study also shows the strong motivation of this paper.

**Reasons To Reject:**

Good paper, I have no reason to reject.

**Reproducibility:**

4: Could mostly reproduce the results, but there may be some variation because of sample variance or minor variations in their interpretation of the protocol or method.

**Reviewer Confidence:**

3: Pretty sure, but there's a chance I missed something. Although I have a good feel for this area in general, I did not carefully check the paper's details, e.g., the math, experimental design, or novelty.

---

> ### Author Rebuttal · Authors · 2023-08-28
>
> Thank you very much for your encouraging comments!
> We hope that our study could inspire further exploration into the reasons and mechanisms behind multilinguality and methods to enhance multilingual pre-training models.
>
> **About the Effectiveness of Our Method on Other mPLMs**
>
> Our study demonstrates that our method would benefit XLM-R (Transformer Encoder) and MASS (Transformer Encoder-Decoder) models. Future work would adapt our method to X-GLM to investigate the effectiveness on the Transformer Decoder based mPLMs.
>
>
>
> We would express our gratitude again for your acknowledgment of our work!

---

### Meta-Review · Area_Chair_ts1n · 2023-09-18

**Recommendation:** 3

**Metareview:**

We thank the author for their insightful paper. This work presents a dictionary-based mask-language modeling (MLMs) pretraining that improves cross-lingual transfer. It first motivates the training method based on a careful analysis of the multilingual internal structure of MLMs (e.g. XLM-R). It then introduces TCS and SCS, which expands masked-language modeling based on dictionary-based cross-lingual pairs and on-the-fly cross-lingual mined candidates.

Reasons to Accept
- Clarity of argumentation (R1, R2)
- Extensive empirical evidence that the introduced method leads to better cross-lingual transfer (R1, R2, R3)

Reasons to Reject:
- Novelty of the results: Dictionary-based training for multilingual modeling has been explored in the past (e.g. Cao et al 2020; Wang et al 2022). The paper would benefit from a more detailed discussion and comparison with these papers (as pointed out by R3)

Suggestions:
- Tables 12 and 13 are tough to read without bolding the best scores.
- One of the arguments in the paper is that monolingual corpora do not include any parallel data or embedded text and still XLM-R-like models are able to learn geometrically-aligned representations. However, e.g., Wikipedia is somehow parallel across languages (many documents are shared across languages). Additionally, many documents include code-switched text. The paper would benefit from a careful discussion of this point and how this could also explain the multilingual abilities of multilingual MLMs.

Missing references
- Expanding Pretrained Models to Thousands More Languages via Lexicon-based Adaptation, (Wang et al 2022)
- First Align, then Predict: Understanding the Cross-Lingual Ability of Multilingual BERT,  (Muller et al 2021)

---

### Decision · Program_Chairs · 2023-10-07

**Decision:**

Accept-Findings

**Comment:**

We thank the author for their insightful paper. This work presents a dictionary-based mask-language modeling (MLMs) pretraining that improves cross-lingual transfer. It first motivates the training method based on a careful analysis of the multilingual internal structure of MLMs (e.g. XLM-R). It then introduces TCS and SCS, which expands masked-language modeling based on dictionary-based cross-lingual pairs and on-the-fly cross-lingual mined candidates.

Reasons to Accept
- Clarity of argumentation (R1, R2)
- Extensive empirical evidence that the introduced method leads to better cross-lingual transfer (R1, R2, R3)

Reasons to Reject:
- Novelty of the results: Dictionary-based training for multilingual modeling has been explored in the past (e.g. Cao et al 2020; Wang et al 2022). The paper would benefit from a more detailed discussion and comparison with these papers (as pointed out by R3)

Suggestions:
- Tables 12 and 13 are tough to read without bolding the best scores.
- One of the arguments in the paper is that monolingual corpora do not include any parallel data or embedded text and still XLM-R-like models are able to learn geometrically-aligned representations. However, e.g., Wikipedia is somehow parallel across languages (many documents are shared across languages). Additionally, many documents include code-switched text. The paper would benefit from a careful discussion of this point and how this could also explain the multilingual abilities of multilingual MLMs.

Missing references
- Expanding Pretrained Models to Thousands More Languages via Lexicon-based Adaptation, (Wang et al 2022)
- First Align, then Predict: Understanding the Cross-Lingual Ability of Multilingual BERT,  (Muller et al 2021)